# *"I abandoned my job to look after my baby."* Understanding the unpriced cost of care of a preterm infant: Caregivers' lived experiences

**Flaviah B. Namiiro**[1]*, **Andrew S. Ssemata**[2,3], **Yaser Abdallah**[1], **Fatuma Namusoke**[4]

**1** Department of Pediatrics& Child Health, Makerere University College of Health Sciences, Kampala, Uganda, **2** Department of Psychiatry, Makerere University College of Health Sciences, Kampala, Uganda, **3** Department of Global Health and Development, London School of Hygiene and Tropical Medicine, London, United Kingdom, **4** Department of Obstetrics & Gynecology, Makerere University College of Health Sciences, Kampala, Uganda

* bnflaviah@gmail.com

## Abstract

### Background

Preterm birth is associated with life-long cost implications on the infant, family, health system, and society at large. The costs related to lost productivity at contributions at work during care of preterm infants are difficult to measure. We aimed to explore and document the unpriced costs parents incur following birth of a preterm infant in the first year of life in a low resource setting.

### Methods

Thirty-nine mothers and five fathers of preterm infants who had ever attended the preterm follow-up clinic after discharge from Mulago National Referral Hospital, were included in a qualitative study between November 2019 and February 2020. Participants were purposively selected, and data were collected using four focused group discussions with mothers and in-depth interviews with the fathers lasting 30–70 minutes each. These were audio-recorded, transcribed and translated. The data were manually analysed using the thematic approach.

### Findings

Three themes were generated: i) complex nature of the infant, ii) time to care for the infant, iii) mother as the predominant caregiver. The parents perceived preterm infants as delicate, complicated and their care more costly compared to those born at term. Expressions of need for time to care for their infants, frequent hospital visits and readmission were raised. Availability of the mother as the predominant caregiver some of whose roles cannot be delegated and their experiences following return to work after birth of a preterm were cited by the participants.

**Data Availability Statement:** All relevant data are within the paper and its Supporting Information files.

**Funding:** FBN received the award Support was from the Forgaty International Center of National Institutes of Health, U.S Department of State's Office of Global AIDS Coordinator and Health Diplomacy (S/GAC), and President's Emergency Plan for AIDS Relief (PEPFAR) under Award Number 1R25TW011213. The content is solely the responsibility of the authors and does not necessarily represent the official views of the National Institutes of Health. The funders had no role in study design, data collection and analysis, decision to publish, or preparation of the manuscript.

**Competing interests:** The authors have declared that no competing interests exist.

## Conclusion

The results highlight the unpriced costs incurred by the parents through disruption of the work pattern due to the actual and perceived needs of a preterm infant and time to care in a low resource setting. We recommend guidance on financial planning, development of policies and programs on social and financial support for parents and future studies on indirect costs of preterm care.

## Introduction

Prematurity is defined by the World Health Organization (WHO) as birth before 37 completed weeks of gestation. These babies are further categorized as being extremely premature if < 28 weeks, moderately or very premature if born 28–32 weeks and late or near premature if born > 32 weeks. These babies may be of low, very low or extremely birth weight < 2500g, < 1500g or <1000g respectively.

Annually, approximately 15 million babies are born prematurely, majority of whom are born in low resource countries (LRCs) [1,2]. In Uganda, about 226, 000 babies are born prematurely every year [3]. Although the reason for premature birth is unknown for the majority of cases, there some associated factors which include; extremes of maternal ages (too young or too old), short birth intervals, no antenatal follow-up, multiple pregnancy, previous preterm birth, infections and illnesses like hypertension [4]. Babies born prematurely are at high risk of mortality as well as morbidity. It is estimated that one million babies die from complications of prematurity annually. Efforts to reduce preterm birth and its complications are ongoing, the majority are health facility centered and most of these babies will have prolonged stay in the facility [5–7].

The cost of care of babies born premature is estimated to be more than 10 times that incurred to care for term babies [8,9]. Literature on care and economic implications following preterm birth in our setting is scarce. The available literature is mainly from the developed countries, mostly on direct costs which are health care related and focused on the health providers' perceptions [10–13].

The direct cost of care has a monetary value attached in form of itemized and or unit cost. On the contrary, the indirect cost of care has no monetary value tagged to it because it is hard to measure and thus unpriced. The unpriced cost of preterm care is related to lost productivity to contributions at work by the parents in the short-term and then the reduced alternatives to work in the long-term [8,14].

Regarding the labor laws of Uganda, there is no mention of support to the family as part of a work incentive following preterm birth and other complex pregnancy outcomes. Affected families in the workforce are left to individual workplace decisions which are sometimes unfavorable. Beyond medical services, there is lack of social support for families with premature infants yet care of such babies stretches beyond medical care. The aim of the study was to explore and document the unpriced cost of care parents incur following birth of a preterm infant for the first year of life in Kampala, Uganda. The findings will help in developing policies and programs on the financial and social support for parents with premature infant(s). **Methods**

### Study design and setting

We conducted an exploratory qualitative study to document parents' views on their experiences regarding the unpriced cost of care for children born preterm in the first year of life. Our

study used a descriptive phenomenology methodological approach as we sought to describe caretakers' experiences, and feelings in response to the 'sacrifices' they made for their preterm babies that cannot be costed. The design was deemed appropriate considering that little is known about the cost of care and their implications in the Ugandan context.

We gathered in-depth information and perceptions using interviews and focused group discussions. From the perspective of our research participants, we report on what this means and how this lived experience affects their caretaking responsibilities and behaviors. Qualitative inquiry facilitates understanding of social processes from the perspectives of the study participants, in this case the parents informed by their lived experiences [15–17]. We undertook a thematic analysis approach to identify themes that were generalized in relation to how the caregivers perceived and experienced taking care of preterm babies with a focus on what we attributed as the unpriced cost.

The study was conducted between November 2019 to February 2020 at the preterm follow-up clinic of Mulago National Referral Hospital, Kampala. Mulago Hospital serves the urban and peri-urban population. It serves as a training institution for Makerere College of Health Sciences and conducts approximately 32,000 births annually (2017 Hospital records) with 13% of them being low birth weight. At birth, the preterm and low birth weight neonates are admitted to the neonatal unit for observation/ treatment and discharged when stable. All neonates born preterm (<37 weeks) and with weight less than 2,500 grams are eligible for follow-up at the clinic after discharge from the neonatal unit. On average, 200 neonates are admitted to the neonatal unit every month and nearly half are low birth weight/preterm. About 50 children are seen in the preterm clinic every week where growth monitoring and counselling on infant feeding are conducted as well as addressing caregivers' concerns. The clinics run twice a week and all professional services are free of charge.

## Participants and recruitment

Mothers and fathers of preterm children who ever attended the preterm follow-up clinic were involved in the study. Parents of children who weighed ≤ 2000 grams at birth and were aged 22–38 months, chronological age at the time of the study were purposively selected from the clinic registry and invited to participate. Parents whose children were sick or admitted at the time of the study, were excluded. The parents were contacted on phone and those who agreed to participate were given an appointment for a face-to-face interview or group discussion at the clinic premises. Participants were included in the study procedures (FGDs and IDIs) after obtaining written informed consent (including permission to audio record the interview or discussion). Mothers participated in the FGDs and the fathers in the IDIs. Mothers are the primary care givers and for uniformity were used for FGDs, and for any unresolved issues we used IDIs with the fathers who are not readily available in the follow-up clinic.

## Data collection procedure

The four FGDs constituting of 8–10 mothers and five IDIs with fathers of preterm children were conducted face-to-face at the follow-up clinic to enlist collective views on cost of care of preterm infants from the parents. Participants were assured of anonymity and confidentiality of their information.

The IDIs and FGDs were conducted by two female research assistants (not employees of the hospital). They were trained and experienced in qualitative research, competent in English and Luganda (local language) and at the time of the study were part of another study where they were performing the same roles as research assistants. None of the authors participated in conducting the interviews to minimize bias since they are involved in the day-to-day activities

of patient care. Discussions were held with the PI (FBN) and the research assistants after each interview to capture any new issues for further exploration.

Before the discussions and interviews started, rapport with the participants was established. Self-introductions to establish rapport with the participants and RAs were made. Introduction of the study objectives and activities were explained to the participants.

During the FGDs and IDIs, one research assistant was the facilitator and the other a note-taker. The sessions were conducted using a guide with open ended questions to allow for flexibility given the sensitivity of the topic under study. The guide was not pilot tested but prompts were added when new issues arose (S1 Appendix). Data saturation was reached after the fourth group discussion as no new information was being generated. The discussions lasted 30–70 minutes, were recorded and transcribed. All audio-recordings from the IDIs and FGDs were transcribed verbatim in Microsoft Word. Those conducted in Luganda were translated into English by another member of the study team.

## Data analysis

Data analysis was conducted manually following a thematic analysis approach. We used inductive coding following reflective thematic analysis by creating themes at a later stage of the analysis. Three stage-thematic synthesis involving the coding of text line-by- line; developing of descriptive themes; and the generation of analytical themes was used. Authors FBN and ASS read the transcripts repeatedly to become familiar with the data, developing initial codes of interest. We generated initial or preliminary codes to our data focusing on the concept of the unpriced costs caregivers narrated to describe the content. We then searched for patterns and themes in our codes across the different interviews which we collated. These we reviewed as a team to ensure we defined and named the themes appropriately. We merged themes that are similar and removed those that did not have enough data to back them up.

Using open coding, patterns were identified, compared, and contrasted meaningful content was condensed and assigned shorter codes to describe it. Subsequently a codebook was developed in Microsoft Excel to support this process. Input was sought from other members of the research team (AY, FN) with extensive experience of obstetric and preterm care to assist with affirming the coding framework. These meetings also helped to ensure that the interpretation of the data was close to the content, drawing on the combined insights and further supported reflectivity (examining our judgement on clinical and non-clinical backgrounds) on the analysis process and the emerging findings. We eventually wrote up our narrative based on the data.

## Ethical considerations

The study was approved by Makerere University School of Medicine Research and Ethics Committee (REC REF 2019–117) and the Uganda National Council of Science and Technology (REF: HS 2690). Administrative clearance was obtained from Mulago Hospital and written informed consent was obtained from the parents before inclusion in the study.

## Results

### Demographic characteristics

A total of 39 mothers participated in the study, most were aged 25–34 years and were married. Majority had secondary education and lived within a 15 km radius from the hospital. More than half of the mothers were in some form of employment, mostly in the informal sector. More than half of the mothers had three or more children. All the fathers were married,

**Table 1. Socio-demographic characteristics for the study participants.**

| Variable | Frequency (N = 39) | Percentage % |
|---|---|---|
| Age | | |
| ≤25 years | 04 | 10.3 |
| 26–34 years | 28 | 71.8 |
| ≥35 years | 07 | 17.9 |
| Estimated Distance from home to hospital | | |
| ≤10Km | 21 | 53.8 |
| 10km | 18 | 46.2 |
| Education level | | |
| Primary below/none | 17 | 43.6 |
| Secondary/tertiary level | 22 | 56.4 |
| Marital status | | |
| Married / co-habiting | 33 | 84.6 |
| Single / divorced/ separated | 6 | 15.4 |
| Employment status | | |
| Employed | 29 | 74.4 |
| Not employed | 10 | 25.6 |
| Number of children | | |
| 1–2 children | 18 | 46.2 |
| ≥3 children | 21 | 53.8 |
| **Fathers' demographics** | | **N = 5** |
| Age ≤ 30 years | 02 | 40.0 |
| > 30 years | 03 | 60.0 |
| Marital status Yes | 05 | 100.0 |
| Education level None/Primary | 01 | 20.0 |
| Upper secondary/Tertiary | 04 | 80.0 |
| Employment status Yes | 05 | 100.0 |

employed, and had some form of education. Summary of the parents' characteristics are shown in Table 1.

## Themes in care of preterm infants in the first year of life

Three major themes emerged on unpriced costs of care for preterm infants in the first year of life and these were: i) complex nature of the infant, ii) time to care for the infant and iii) mother as the predominant caregiver. The themes and considerations are presented in Table 2.

**Table 2. Emerging themes and subthemes on unpriced costs of preterm infants.**

| Major themes | Subthemes |
|---|---|
| i) Complex nature of the infant | • Preterm infants' small size<br>• Infants are sickly and prone to infections<br>• Infants grow slowly |
| ii) Time to care for the infant | • Need for prolonged hospitalization<br>• Frequent hospital visits and readmission<br>• Need for follow-up |
| iii) Mother as the predominant caregiver | • Mothers' availability to provide and care for infant<br>• Mother working for a living and her nature of work |

Theme 1: Complex nature of the infant

Nearly all parents cited that the babies born preterm were delicate and complicated compared to those born at term. Being delicate was related to the babies being small and not weighing as much as a term baby. They were also described as being sickly because they were prone to infections. The parents stated that the babies did not grow well by gaining weight due to their poor health. This meant one had to be careful when taking care of them. Some of the parents who stated having no problem taking care of the preterm babies were first time mothers who had no prior experience. Others reported following the instructions given to them by the health workers.

*"These babies are delicate, and we made sure that the environment is perfectly clean to avoid infections. When we followed the instructions given by health workers, the baby never had any complications apart from the usual flue." FGD 2.*

*"They (preterm babies) are complicated and require a lot of commitment." IDI 1*

*"The baby was very sickly; he would not grow fat, so I came to hospital continuously." FGD 1.*

Theme 2: Time to care for the infant,

Most parents stated the babies require a lot of time and attention compared to those born at term. Infants born preterm need to be fed every two to three hours either using a tube or cup as they cannot breastfeed well. They should be monitored continuously, provide them with warmth by the Kangaroo method and all these require time. The parents stated that these babies were at higher risk of ill health and stayed longer in hospital. They required more re-hospitalization and frequent hospital visits for follow-up compared to term infants. Some parents perceived the follow-up visits positively, but all these were expensive, and one needed a lot of time.

*"It is always good to bring the baby for follow up; as for my wife, this was her first-born baby and she had no idea of what takes place during such challenging moments. The follow-ups helped us to avoid making mistakes which could affect the baby." IDI 4.*

*"Children who are born preterm need a lot of time compared to those who are born at term. Their brains are not fully developed so you have to ensure that you give them a lot of attention." FGD 3.*

Some participants reported that other key caregivers invested time and efforts in caring for the baby while they supported the mother. They pointed out that some of the babies required a lot of care to follow the health workers' recommendations and mothers too need the support from their relatives.

*"My mother in-law forced me to bring the baby in the hospital when I came the health worker looked at the feeding tube which was looking very unhealthy and the baby was in the bad shape, she counseled me. I started to follow the instructions, gave the baby a lot of care and he started growing slowly. He was too tiny, I used to refer to him as "Kameese" (translated small rat). Me and my sister would feed him through the tube until he grew big, although it was not easy. Now we named my baby Prince "Kimeese" (translated big rat)." FGD 3.*

Theme 3: Mother as the predominant caregiver

Mothers were perceived as the best caregivers for their babies compared to other people. Their availability to provide care for the babies needed them to be fully committed to this duty

if they expected the best outcome for them. This role they started at birth, during admission of the baby and after discharge then continue to carry on with their other obligations. Some mothers expressed their need for help from especially their husbands because it was difficulty to handle on their own.

> "*What we the parents need to do, first and foremost there must be cooperation between the husband and wife. For example, the wife may be busy doing something else and the child needs attention, the man should help the wife because the child doesn't belong to only one person.*" FGD 2.

> "*First of all, we have our jobs, other children we have left home and our marriages. We also have to be in the hospital to take care of this child, this tortures us psychologically yet even this child needs your complete attention.*" FGD 3.

The mothers who worked for a living and or worked away from home had different experiences regarding their employment and having to take care of their babies. For example, those who were not self-employed had to make decisions which either affected them or their infants. Some reported having to stop work to take care of their babies. A few mothers had flexible options which favored them and their infants, but this was not the case for others. They highlighted the extra costs they had to incur during care for their infants such as getting other people to help them.

> "*It depends on the job one has but for me they allowed me to go to work with my baby. These children are really expensive because their visits to the hospital are endless. But if your child starts sitting you can go with them to work and put them to sit as you work.*" FGD 4.

Some mothers noted that since the labor starts unexpectedly and prematurely, they would have to stop everything they were been doing without a plan including having to stop work or lose their jobs when they stayed away for long.

> "*I put everything aside although I was working, I abandoned my job for the baby.*" FGD 2.

> "*The challenge we have is that after we deliver these children, we leave them in the hands of maids to give them expressed milk. The maids will leave these children to sleep which causes most of these preterm children to grow at a slow rate.*" FGD 1.

Some mothers will incur costs to do anything they think will benefit their infants irrespective of the importance.

> "*For me I bought a weighing scale for my child (joint laughter). I kept weighing her to ensure that she had more weight than the last time I weighed her.*" FGD 1.

## Discussion

We explored the lived experiences of parents to children born preterm to understand the unpriced costs of care parents incur during the first year of life. Three themes emerged from the discussions: complex nature of the infant, time to care for the infant and mother as the predominant caregiver.

The complex nature of the infant could have been viewed by majority of the parents as leverage for their infants' survival. This implied whatever should be done to save the infant

should be done irrespective of the cost. Infants' need for long healthcare during initial hospitalization and after discharge causes anxiety for fear of poor outcomes. Parents' actions during care of their infant are sometimes taken with caution to avoid making mistakes, however, some decisions lead to large hidden costs that are unknown to the parents. Unlike their term counterparts, preterm infants require more time for their day-to-day care both during admission and after discharge, especially during the first year of life and the mother plays a central role.

The small, repetitive lifesaving roles caregivers perform for their infants, such as the two hourly feeds and kangaroo care, require time and commitment. These may not be easily delegated to a person who is not a nuclear family member or has no prior experience of taking care of a preterm infant. Also, the bond the parents developed during the critical period explains the reason not to trust another person with the comprehensive care a baby requires. Hence prioritizing care for their infant, although this may disrupt their life.

The time from birth of a preterm infant to the time they are declared out of danger varies from infant to infant. Similarly, the level of stress, amount of time, resources, and effort put into caring for an infant varies between families. Therefore, attaching a cost to these individual aspects of care is difficult because estimating them is hard. Henceforth, it is not surprising the parents viewed cost of care of preterm infants as expensive since their input is often different. In a previous study, health providers found that the cost of care of infants with lower gestation age and birth weight was higher [14,18]. Physical and emotional drains which may affect performance that were observed in this study have also been reported elsewhere [19,20].

Furthermore, when the mother must delegate her duties to another person for instance, when resuming work, there was no guarantee that the delegated caregiver would perform exactly as the mother and vice versa. Similar findings are lacking from literature, hence, the need for more studies in this area. In a nutshell, mothers remain the best caregivers for their preterm babies. Also, flexibility at the workplace or absenteeism to the extremes of having to stop work, all disrupt the life of an individual, their family and community at large.

Our study findings indicate care of a preterm infant is generally more costly than care of a term baby which is consistent with other studies [9,11]. Having time to care for the preterm was the most important unpriced driver of cost of care. The mothers contribute more to the care of the preterm infants and their availability was very crucial. Available literature on economic impact is mostly based on the health providers' perspective and from high income settings [7,13,21]. This study highlighted the parents' views and focused on the indirect cost of care. Tagging a price to most of the intangible costs such as time invested in care of the preterm, stress and anxiety, disruption of work and contribution from other members of the family is difficult.

The findings indicate that the unpriced cost of care of preterm infants are beyond financial costs. Caregivers rarely have an idea of this expenditure following preterm birth and its impact on the rest of the family resources. For many families, there is no time for financial planning since the birth happens when its least expected. Their decision on duration of hospital stay, likely expenditure during and after admission, and when mother is likely to return to work is limited. Parents need continued support during admission, prior to discharge and during the follow-up visits to understand these "hidden" costs to plan for infant care with minimal family disruptions.

Although these infants are at risk of long-term health complications the parenting process can be made less stressful and affordable for each family with continued support and stepwise planning. Currently there are no known work policies both in the formal and informal sectors to support workers with high-risk births such as those born preterm, during and after hospital discharge. The entire burden of care is shifted to the family, and this subsequently affects the

infants' outcome, community, and contribution to the workforce. We recommend for special consideration of policies like extended maternity leave for high-risk infants to increase mothers' availability to care for their infants and state financial support to the family, and more research on this topic in low resource settings.

## Strengths and limitations of the study

The findings of this study should be viewed considering these strengths and limitations. The explorative nature of the study provides understating of the unpriced costs caregivers experience in context. This is one of the few studies to address the topic on cost in our setting. The findings provide critical information for further research to understand the indirect costs parents incur and guide policy development to support them. The limitations of this study include being a single center study and not handling caregivers in same groups of socioeconomic strata: high versus low. However, Mulago hospital being a national referral center receives individuals from across the country and from different socioeconomic status.

## Conclusion

We highlight the unpriced costs incurred by the parents through disruption of the work pattern due to the actual and perceived needs of a preterm infant and time to care in a low resource setting. We recommend guidance on financial planning for parents, development of policies, and programs on social, and financial support and future studies on the topic of indirect costs of preterm care.

## Supporting information

**S1 Appendix. Interview guide.**
(DOCX)

## Acknowledgments

We are particularly grateful to the study participants, research assistants and the staff at the preterm clinic at Mulago national referral hospital.

## Author Contributions

**Conceptualization:** Flaviah B. Namiiro, Yaser Abdallah, Fatuma Namusoke.

**Data curation:** Flaviah B. Namiiro.

**Formal analysis:** Flaviah B. Namiiro, Andrew S. Ssemata.

**Funding acquisition:** Flaviah B. Namiiro.

**Methodology:** Flaviah B. Namiiro, Yaser Abdallah, Fatuma Namusoke.

**Supervision:** Flaviah B. Namiiro, Andrew S. Ssemata.

**Validation:** Fatuma Namusoke.

**Writing – original draft:** Flaviah B. Namiiro, Andrew S. Ssemata.

**Writing – review & editing:** Flaviah B. Namiiro, Andrew S. Ssemata, Yaser Abdallah, Fatuma Namusoke.

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
