## [Decision Letter · Decision Letter 0]

27 Feb 2023

PONE-D-22-33576“I abandoned my job to look after my baby.” The unpriced cost of taking care of a preterm infant: caregivers’ lived experiences.PLOS ONE

Dear Dr. Namiiro,

Thank you for submitting your manuscript to PLOS ONE. After careful consideration, we feel that it has merit but does not fully meet PLOS ONE’s publication criteria as it currently stands. Therefore, we invite you to submit a revised version of the manuscript that addresses the points raised during the review process.

We look forward to receiving your revised manuscript.

Kind regards,

Milton W. Musaba, M.D

Academic Editor

PLOS ONE

Journal Requirements:

"FBN received the award

Support was from the Forgaty International Center of National Institutes of Health, U.S Department of State’s Office of Global AIDS Coordinator and Health Diplomacy (S/GAC), and President’s Emergency Plan for AIDS Relief (PEPFAR) under Award Number 1R25TW011213. The content is solely the responsibility of the authors and does not necessarily represent the official views of the National Institutes of Health."           

Additional Editor Comments:

Dear Dr. Flavia Namiiro

Thank you for submitting your manuscript to this journal.

After careful consideration of your manuscript, I agree with the reviewer that it requires Major revision.

Please see the comments for your consideration.

Reviewer 1;

Thank you for focusing your research on the pertinent area concerned with the care of pre-term infants. It highlights the hidden experiences faced by the caretakers of these children.

The document is written in a simple way with adherence to the qualitative research standards, however, a number of areas need clarification and improvement as noted below.

Abstract

The abstract is concise with a good introduction, methodology and results.

Sentences 15 to 16; the finding is not clear and doesn’t bring out the participant’s views about the mother of the pre-term as a caregiver. You need to make it clear.

In conclusion sentences 19 to 21, the recommendation is good however, it mainly favours women in formal employment. What about those in informal employment especially those employing themselves who equally incur such costs? What is your thought about them?

General thoughts about the introduction:

The introduction is well written with some good background information, statistics and the consequences for pre-term births in a society, however, you need to position your study in your context as well. For instance, how many babies are born preterm annually or what is the incidence?

Then in this section also, wouldn’t it be good to briefly mention the factors associated with pre-term birth, and its prevention in a sentence or two, followed by the consequences?

Are there studies that have looked at the same topic from a high-income and/or a low-income setting? What did they find?

You may want to define a pre-term birth in terms of gestation age and birth weight so that non-medical audiences clearly understand the group of your focus.

Sentence 32, I guess that is 32 weeks, if so, you may want to indicate that it is “32 weeks”

In sentences 31 to 43, you may want to organize the sentences in a way that you define what the indirect cost is versus direct cost, how indirect costs come about, how it is measured and what makes it difficult to estimate the indirect cost in your setting despite its importance.

Sentences 48 to 49, Data on the estimate of cost incurred".... this is not clear. Did you set out to estimate the financial cost? You may want to be consistent on the phrase “indirect costs” throughout the text.

The gap on which this study is based is not clear..., you may want to state it clearly.

Note, amidst all these, you may want to maintain the introduction section brief despite the information you are likely to add. So, in this section, it is important that the reader is given background information as the argument is built amidst justifying the need for the study. All these need to have a chronological flow until the existing gap and how the study will fill the gap. Don’t forget to mention how your findings will be useful in a sentence also.

Methods

What is the design used in the study?

Your study could benefit from a methodological orientation since you are exploring the views or experiences of the caretakers of pre-term babies. For instance, phenomenology etc

Sentence 101; what was the main objective? was it the cost of care? you need to define the "cost of care"

Sentence 106; What questions did the guide have? did you pre-test it? You don’t talk much about the IDIs, could there have been more information to give on the IDIs.

Sentence 106; most times, the “sensitive information” are collected using in-depth interviews. However, you had a few IDIs conducted.

It is also important to give the rationale for choosing women to participate in FGD and men in IDIs.

Was saturation realized with IDIs? Or Did FGD serve as the main data collection method?

Who were the data collectors, and was there any rapport created between the research assistants and participants before the interviews?

Data analysis

Sentences 114 to 119; During the analysis, did you have apriori themes?

In other words, was this analysis deductive or inductive? Or were themes identified in advance or derived from the data?

Stemming from your objective, what exactly did you do after translating and familiarizing yourself with the data?

Organizing the steps, you did during the analysis would enable the reader to clearly understand the analysis steps.

Are the considerations categories or sub-themes?

You need to explain them in the results, apparently, it doesn’t come out well in the results. or you could put some small sub-headings as you explain the “considerations”.

Kindly write about the study team and reflexivity. This is very important for the reader to know the perspective from which the study was conducted. It adds to the reliability and validity of the findings. You can be guided by COREQ (consolidated criteria for reporting qualitative research)

Sentence 121, second word; what is the meaning of the term as applied to your data?

Results section

Well written however, a few comments

Table 2, last row, second column: what is the difference between the 2 considerations “mother worked for a living” and “mother’s nature of work and where they work”

Sentence 163, is that word health or healthy

Discussion

The discussion is mainly full of the findings re-stated with little or no interpretation.. I think it would be good to state the finding briefly, give its explanation or interpretation or meaning and the implication on the baby or mother or family etc and relate to a previous study.

Objective

Sentences 233 to 234; This is similar to your objective, but not the same.

"Explore and document the indirect costs to parents towards the care of a preterm infant in the first year of life in a low resource setting" this is what you have in the abstract.

Then in the introduction

“The aim of the study was to explore indirect costs parents incur following birth of a preterm infant and document the parents’ experiences on care in the first year of life”

Then in the method section

“We conducted an exploratory qualitative study to explore parents’ views on their experiences regarding the unpriced cost of care of children born preterm and with low birth weight”

Based on these, First, you want to explore the indirect costs associated with preterm births, secondly, you wanted to document the parent’s experiences on care in the first year.

I think it’s better for you to decide on the message you want to communicate in relation to pre-term baby care and clearly state the objective and be consistent across the different sections.

Sentences 237 to 238; I found it a good statement that could add up to the argument or gap in the introduction section.

References

You need to use the writing style recommended by PLOS ONE for your references and in-text citation. Kindly refer to Author’s instructions or guide.

Reviewer 2

Abstract:

Line 3: what other costs are you referring to, if they are not direct

Line 12 -13: i) complex nature of the infant, ii) time to care for the infant, iii) mother as the primary caregiver. The phrasing of these findings, does not make them sound like indirect costs that you set out to explore.

Introduction:

This section is not focused on the challenge at hand – indirect costs associated with care for newborns after discharge from the hospital.

1. The problem/gap in knowledge is not clearly articulated in this section.

2. What have other researchers reported on this subject of indirect costs in low resource settings, beyond mentioning that it is estimated to be 10 times higher than in hospital care.

3. Please revise this to fous on; what do we know about this issue, what don’t we know about it(gap)?, and what do you want to do about it? (aim of your study)

Methods:

One of the key things missing in this section is the lack of a conceptual/theoretical frame work that has informed this analysis. Relatedly, the Equator guidelines for reporting qualitative research were not followed, this makes this section very uninformative and missing a lot of relevant information.

Results;

In table 1- please provide an informative and descriptive title that can make it stand alone.

– complete the table margins as well.

Discussion

1. The key message in this section is difficult to discern in the current state, it is probably related to the misalignment between the title, aim and results reported. What should be the take home message for the readers?

2. What theoretical framework underpins the discussion of your results?

3. Line 280 – it should be cost and not course. Relatedly, there are many typos and grammatical errors all over the document that you need to revise.

Conclusion:

Line 295- “provides understating of the unpriced “hidden” costs caregivers experience in context”. I believe that this wording might be a better way to state your aim and title of the study given the results being reported, as opposed to “indirect costs”.

Reference:

1. Number 6 – 11 are incomplete

2. Please check al the other references to ensure that they conform to the journal guidelines.

Reviewers' comments:

Reviewer's Responses to Questions

**Comments to the Author**

1. Is the manuscript technically sound, and do the data support the conclusions?

Reviewer #1: Yes

2. Has the statistical analysis been performed appropriately and rigorously? 

Reviewer #1: N/A

3. Have the authors made all data underlying the findings in their manuscript fully available?

Reviewer #1: Yes

4. Is the manuscript presented in an intelligible fashion and written in standard English?

Reviewer #1: Yes

5. Review Comments to the Author

Reviewer #1: Dear authors,

Thank you for focusing your research on the pertinent area concerned with the care of pre-term infants. It highlights the hidden experiences faced by the caretakers of these children.

The document is written in a simple way with adherence to the qualitative research standards, however, a number of areas need clarification and improvement as noted below.

Abstract

The abstract is concise with a good introduction, methodology and results.

Sentences 15 to 16; the finding is not clear and doesn’t bring out the participant’s views about the mother of the pre-term as a caregiver. You need to make it clear.

In conclusion sentences 19 to 21, the recommendation is good however, it mainly favours women in formal employment. What about those in informal employment especially those employing themselves who equally incur such costs? What is your thought about them?

General thoughts about the introduction:

The introduction is well written with some good background information, statistics and the consequences for pre-term births in a society, however, you need to position your study in your context as well. For instance, how many babies are born preterm annually or what is the incidence?

Then in this section also, wouldn’t it be good to briefly mention the factors associated with pre-term birth, and its prevention in a sentence or two, followed by the consequences?

Are there studies that have looked at the same topic from a high-income and/or a low-income setting? What did they find?

You may want to define a pre-term birth in terms of gestation age and birth weight so that non-medical audiences clearly understand the group of your focus.

Sentence 32, I guess that is 32 weeks, if so, you may want to indicate that it is “32 weeks”

In sentences 31 to 43, you may want to organize the sentences in a way that you define what the indirect cost is versus direct cost, how indirect costs come about, how it is measured and what makes it difficult to estimate the indirect cost in your setting despite its importance.

Sentences 48 to 49, Data on the estimate of cost incurred".... this is not clear. Did you set out to estimate the financial cost? You may want to be consistent on the phrase “indirect costs” throughout the text.

The gap on which this study is based is not clear..., you may want to state it clearly.

Note, amidst all these, you may want to maintain the introduction section brief despite the information you are likely to add. So, in this section, it is important that the reader is given background information as the argument is built amidst justifying the need for the study. All these need to have a chronological flow until the existing gap and how the study will fill the gap. Don’t forget to mention how your findings will be useful in a sentence also.

Methods

What is the design used in the study?

Your study could benefit from a methodological orientation since you are exploring the views or experiences of the caretakers of pre-term babies. For instance, phenomenology etc

Sentence 101; what was the main objective? was it the cost of care? you need to define the "cost of care"

Sentence 106; What questions did the guide have? did you pre-test it? You don’t talk much about the IDIs, could there have been more information to give on the IDIs.

Sentence 106; most times, the “sensitive information” are collected using in-depth interviews. However, you had a few IDIs conducted.

It is also important to give the rationale for choosing women to participate in FGD and men in IDIs.

Was saturation realized with IDIs? Or Did FGD serve as the main data collection method?

Who were the data collectors, and was there any rapport created between the research assistants and participants before the interviews?

Data analysis

Sentences 114 to 119; During the analysis, did you have apriori themes?

In other words, was this analysis deductive or inductive? Or were themes identified in advance or derived from the data?

Stemming from your objective, what exactly did you do after translating and familiarizing yourself with the data?

Organizing the steps, you did during the analysis would enable the reader to clearly understand the analysis steps.

Are the considerations categories or sub-themes?

You need to explain them in the results, apparently, it doesn’t come out well in the results. or you could put some small sub-headings as you explain the “considerations”.

Kindly write about the study team and reflexivity. This is very important for the reader to know the perspective from which the study was conducted. It adds to the reliability and validity of the findings. You can be guided by COREQ (consolidated criteria for reporting qualitative research)

Sentence 121, second word; what is the meaning of the term as applied to your data?

Results section

Well written however, a few comments

Table 2, last row, second column: what is the difference between the 2 considerations “mother worked for a living” and “mother’s nature of work and where they work”

Sentence 163, is that word health or healthy

Discussion

The discussion is mainly full of the findings re-stated with little or no interpretation.. I think it would be good to state the finding briefly, give its explanation or interpretation or meaning and the implication on the baby or mother or family etc and relate to a previous study.

Objective

Sentences 233 to 234; This is similar to your objective, but not the same.

"Explore and document the indirect costs to parents towards the care of a preterm infant in the first year of life in a low resource setting" this is what you have in the abstract.

Then in the introduction

“The aim of the study was to explore indirect costs parents incur following birth of a preterm infant and document the parents’ experiences on care in the first year of life”

Then in the method section

“We conducted an exploratory qualitative study to explore parents’ views on their experiences regarding the unpriced cost of care of children born preterm and with low birth weight”

Based on these, First, you want to explore the indirect costs associated with preterm births, secondly, you wanted to document the parent’s experiences on care in the first year.

I think it’s better for you to decide on the message you want to communicate in relation to pre-term baby care and clearly state the objective and be consistent across the different sections.

Sentences 237 to 238; I found it a good statement that could add up to the argument or gap in the introduction section.

References

You need to use the writing style recommended by PLOS ONE for your references and in-text citation. Kindly refer to Author’s instructions or guide.

6. PLOS authors have the option to publish the peer review history of their article (what does this mean?). If published, this will include your full peer review and any attached files.

Reviewer #1: No

---

## [Author Response · Author response to Decision Letter 0]

28 Apr 2023

12 April 2023

Dear Editor,

Thank you to the reviewers for the feedback and comments to strengthen our manuscript. Below, we have addressed the comments point-by-point, attached a copy of the manuscript with track changes and a clean copy. 

Reviewer # 1 Thank you for focusing your research on the pertinent area concerned with the care of pre-term infants. It highlights the hidden experiences faced by the caretakers of these children.

The document is written in a simple way with adherence to the qualitative research standards, however, a number of areas need clarification and improvement as noted below.

Response: Thanks to the reviewer for the kind comment and highlighting the clarifications and the improvements needed to make our manuscript better.

Abstract

The abstract is concise with a good introduction, methodology and results.

Response: Thank you for the feedback.

1) Sentences 15 to 16; the finding is not clear and doesn’t bring out the participant’s views about the mother of the pre-term as a caregiver. You need to make it clear.

Response: Thank you for pointing out this unclarity. 

This view is to state: The mother as the predominant caregiver and may have some roles she may not delegate. 

To clarify, the sentence: “Availability of a primary caregiver especially the mother and their experiences following return to work after birth of a preterm infant were highlighted by the participants.” 

Has been rewritten to read as follows: 

“Availability of the mother as the predominant caregiver some of whose roles cannot be delegated and their experiences following return to work after birth of a preterm infant were cited by the participants. 

Page: l, Lines: 15-17

2) In conclusion sentences 19 to 21, the recommendation is good however, it mainly favours women in formal employment. What about those in informal employment especially those employing themselves who equally incur such costs? What is your thought about them?

Response: Thank you for the comment.

Yes, the recommendation seems to favour those in formal employment and not those in the informal employment. Our thought about those in informal employment is that they manage their time and make decisions regarding their work and yet continue to care for their preterm infants. This flexibility may not be readily available for the women in formal employment in our context.

We have however restated our recommendation so that those in the formal employment benefit and reads as follow:

We recommend a longer maternity leave, family counselling on resource planning, social and financial support and future studies looking at indirect costs and how to mitigate them following preterm birth. 

Page: l, Lines: 19 -21

 3) General thoughts about the introduction: 

The introduction is well written with some good background information, statistics and the consequences for pre-term births in a society, however, you need to position your study in your context as well. 

a. For instance, how many babies are born preterm annually or what is the incidence?

Response: Thank you for asking this question. 

The sentence below has been included in the manuscript.

In Uganda, about 226, 000 babies are born prematurely every year.

Pg: 2, Line: 29

b. Then in this section also, wouldn’t it be good to briefly mention the factors associated with pre-term birth, and its prevention in a sentence or two, followed by the consequences?

Response: Thank you for this important suggestion about mentioning factors associated with preterm birth and its prevention and consequences. 

This paragraph has been improved and now reads as follows: 

Most babies are born prematurely for unknown reasons although some factors have been associated with preterm birth. These include; poverty, extremes of maternal ages (too young or too old), short birth intervals, no antenatal follow-up, multiple pregnancy, previous preterm birth, infections and illnesses like hypertension. Babies born prematurely are at high risk of mortality as well as morbidity. It is estimated annually that one million babies die from complications of prematurity. Efforts to reduce preterm birth and its complications are ongoing, the majority are health facility centered and most of these babies will have prolonged stay in the facility.

Pg 2, Lines 29 - 35

c. Are there studies that have looked at the same topic from a high-income and/or a low-income setting? What did they find?

Response: Thank you for this important question. The few available studies mainly from the high-income countries focus on direct costs. We have not found much literature on indirect costs both in low- and high-income countries, thus the backbone for conducting and writing this research.

d. You may want to define a pre-term birth in terms of gestation age and birth weight so that non-medical audiences clearly understand the group of your focus.

Response: Thank you very much for this suggestion

The definitions of the preterm and birth weight have been added to the manuscript so that non-medical audiences clearly understand the group of focus. They read as follows in the text: 

Prematurity is defined by the World Health Organization (WHO) as birth before 37 completed weeks of gestation. These babies are further categorized to being extremely premature if < 28 weeks, moderately or very premature if born 28 – 32 weeks and late or near premature if born > 32 weeks. These babies may be of low birth weight < 2500g, very low birth weight < 1500g or extremely low birth weight <1000g.

Pg 2, Line 24 – 27

e. Sentence 32, I guess that is 32 weeks, if so, you may want to indicate that it is “32 weeks”

Response: Thank you for pointing out this unclarity. This has been clarified to indicate 32 weeks in the definition above.

Pg 2, Line 25 – 26

f. In sentences 31 to 43, you may want to organize the sentences in a way that you define what the indirect cost is versus direct cost, how indirect costs come about, how it is measured and what makes it difficult to estimate the indirect cost in your setting despite its importance.

Response: Thank you for this important suggestion and guidance. The sentences have been re-written to add the definitions of the direct and indirect costs and now reads as follows:

The direct cost of care has a monetary value attached in form of itemized and or unit cost. The indirect cost of care on the other hand is hard to measure, to have a monetary value tagged to it and hence is unpriced.

Pg 2, Line 40 - 41

g. Sentences 48 to 49, Data on the estimate of cost incurred".... this is not clear. Did you set out to estimate the financial cost? You may want to be consistent on the phrase “indirect costs” throughout the text.

Response: Thank you for pointing out this unclarity. The sentence has been improved to maintain consistence and now reads:

There is scanty literature on cost of care for preterm babies and no literature on the indirect cost of care of preterm infants in LRCs.

Pg 2, Line 38 - 39

h. The gap on which this study is based is not clear..., you may want to state it clearly.

Response: Thank you for pointing out the gap in our study. 

The gap is scarcity of literature on indirect costs incurred by parents taking care of preterm infants in our setting.

This statement has been included in the manuscript.

The cost of care of babies born premature is estimated be more than 10 times that incurred in the care of term babies. These estimated costs are majorly healthcare related costs and literature is mostly from high income countries (HIC). There is scanty literature on cost of care for preterm babies and no literature on the indirect cost of care of preterm infants in LRCs.

Pg 2, Line 36 - 39

i. Note, amidst all these, you may want to maintain the introduction section brief despite the information you are likely to add. So, in this section, it is important that the reader is given background information as the argument is built amidst justifying the need for the study. All these need to have a chronological flow until the existing gap and how the study will fill the gap. 

Response: Thank you for sharing this caution. The introduction section has been re-written and an effort to maintain it brief has been made. Background information has been given while building the argument to justify the need for the study.

Pg 2- 3, Line 24 - 41

j. Don’t forget to mention how your findings will be useful in a sentence also.

Response: Thank you for the advice. 

The findings from the current study will add to the knowledge gap on the topic of unpriced costs incurred by parents during care of a preterm infant. 

This sentence has been added to the introduction section in the manuscript.

Pg 3, Line 52 - 53

4) Methods

i. What is the design used in the study?

Response: Thank you for asking.

The study design was a qualitative design. 

This has been added to the manuscript to clarify.

Pg 4, Line 57 

ii. Your study could benefit from a methodological orientation since you are exploring the views or experiences of the caretakers of pre-term babies. For instance, phenomenology etc

Response: Thank you for the guidance regarding our methodological orientation. This has been improved and reads as follows:

We conducted an exploratory qualitative study to document parents’ views on their experiences regarding the unpriced cost of care of children born preterm in the first year of life. Our study used a descriptive phenomenology methodological approach as we sought to describe caretakers’ experiences, feelings and responses in response to the ‘sacrifices’ they made for their preterm babies that cannot be costed. The design was deemed appropriate considering that little is known about the cost of care and their implications in the Ugandan context. We gathered in-depth information and perceptions using interviews and group discussions. From the perspective of our research participants, we report on what this means and how this lived experience affects their caretaking responsibilities and behaviors. Qualitative inquiry facilitates understanding of social processes from the perspectives of the study participants, in this case the parents informed by their lived experiences. We undertook a thematic analysis approach in order to identify themes that are generalized in relation to how the caregivers perceived and experienced taking care of preterm babies with a focus on what we attribute as the unpriced cost. 

Pg 4, Line 57 - 68

iii. Sentence 101; what was the main objective? was it the cost of care? you need to define the "cost of care"

Response: Thank you for highlighting this point.

The main objective of the study was to explore and document unpriced costs incurred by the parents during care of their preterm infant in the first year of life.

By ‘’cost of care,” we implied “the unpriced costs.” This has been rectified in the manuscript.

Pg 3, Line 50 - 52

iv. Sentence 106; What questions did the guide have? did you pre-test it? You don’t talk much about the IDIs, could there have been more information to give on the IDIs.

Response: Thank you for these important questions.

The questions were related to the parents’ experiences taking care of their preterm infants. The questionnaire was not pretested but discussions continued until there was no new information generated. The guide with questions was added as a supplementary.

Regarding the IDIs, most of the information was similar to that shared from the FGDs, and quotes where captured form both the FGDs and IDIs.

v. Sentence 106; most times, the “sensitive information” are collected using in-depth interviews. However, you had a few IDIs conducted.

Response: Thank you for highlighting this information.

It is true some of the sensitive information was collected using IDIs where fathers felt open to share in-depth personal experiences. In our setting, the mothers are the predominant caregivers for the infants than the fathers. During the analysis, we realized that the views expressed in the FGDs were not different from those from the IDIs. Both caregivers held similar perspectives.

vi. It is also important to give the rationale for choosing women to participate in FGD and men in IDIs.

Response: Thank you for pointing this out.

In our setting, the mothers are more involved in infant care than the fathers, they are the predominant caregivers. We used them in the FGDs for uniformity and for any unresolved issues we used the IDIs with the fathers who were not involved in the FGDs. The fathers were fewer and attaining a reasonable number to conduct a group discussion proved impractical during the conduct of the study. It is imperative to note however that the views expressed during the FGDs were not different from these from the IDIs. 

vii. Was saturation realized with IDIs? Or Did FGD serve as the main data collection method?

Response: Thank you for asking this question.

Yes, saturation was realized with both the IDIs and the FGDs. We considered both IDIs and FGDs as the main data collection methods as each focused on obtaining data from specific group of people in addressing the research question. We analysed and triangulated the data collected from both sources as they held important views on the topic of interest. 

viii. Who were the data collectors, and was there any rapport created between the research assistants and participants before the interviews?

Response: Thank you for asking this important question.

The data collectors were two researchers trained in qualitative research. One lead the discussion and the other was a notetaker. Rapport was created before each discussion was started and a feedback meeting was held with the principal investigator to capture any new ideas from the FGD. This has been clarified in the manuscript.

Pg 5 & 6, Line 94 - 101

5) Data analysis

i. Sentences 114 to 119; During the analysis, did you have apriori themes?

In other words, was this analysis deductive or inductive? Or were themes identified in advance or derived from the data?

Response: Thank you for this important question.

We used inductive coding following reflective thematic analysis by creating themes at a later stage of the analysis. We derived emergent themes, created categories through interactive reading and abstraction from the data collected from the research participants by two authors FBN and ASS as indicated in the manuscript.

This has been clarified in the manuscript. This passage has been added to the manuscript.

We used inductive coding following reflective thematic analysis by creating themes at a later stage of the analysis. Three stage-thematic synthesis involving the coding of text line-by- line; developing of descriptive themes; and the generation of analytical themes was used.

Pg 6, Line 112 - 115

ii. Stemming from your objective, what exactly did you do after translating and familiarizing yourself with the data? Organizing the steps, you did during the analysis would enable the reader to clearly understand the analysis steps.

Response: Thank you for pointing out this clarification. Below is the stepwise analysis we undertook:

We generated initial or preliminary codes to our data focusing on the concept of the unpriced costs caregivers narrated in order to describe the content. We then searched for patterns and themes in our codes across the different interviews which we collated. These we reviewed as a team to ensure we defined and named the themes appropriately. We merged together themes that are similar and removed those that did not have enough data to back them up. Using open coding, patterns were identified, compared and contrasted, meaningful content was condensed and assigned shorter codes to describe it. Subsequently a codebook was developed in Microsoft Excel to support this process. Input was sought from other members of the research team (AY, FN) with extensive experience of obstetric and preterm care to assist with affirming the coding framework. These meetings also helped to ensure that the interpretation of the data was close to the content, drawing on the combined insights and further supported reflectivity (examining our judgement on clinical and non-clinical backgrounds) on the analysis process and the emerging findings. Eventually, we began to write up our narrative based on the data. 

Pg 6-7, Line 116 – 121

iii. Are the considerations categories or sub-themes? You need to explain them in the results, apparently, it doesn’t come out well in the results. or you could put some small sub-headings as you explain the “considerations”.

Response: Thank you for this question.

The key considerations are subthemes and to make it clear for the reader, we have replaced the words key considerations with subthemes. We chose to write up the results as block narratives without subheadings for each subtheme as these ideas were closely related. In order to create a systematic narrative of the discourse, we decided not to break down the results into smaller sections of subthemes but rather write it up based on key themes.

Table 2

iv. Kindly write about the study team and reflexivity. This is very important for the reader to know the perspective from which the study was conducted. It adds to the reliability and validity of the findings. You can be guided by COREQ (consolidated criteria for reporting qualitative research)

Response: Thank you for the guidance. A note on reflexivity has been included in the manuscript and reads as follows:

The IDIs and FGDs were conducted by two female research assistants (not employees of the hospital). They were trained in qualitative research, competent in English and Luganda (local language) and at the time of the study were part of another study where they were performing the same roles as qualitative research assistants. None of the authors participated in conducting the interviews to minimize bias since they are involved in the day-to-day activities of patient care. Discussions were held with the PI (FBN) and the research assistants after each interview to capture any new issues for further exploration. 

Pg 4 - 5, Line 93 -101

vi. Sentence 121, second word; what is the meaning of the term as applied to your data?

Response: Thank you for asking this question. This was used in error and has been excluded from the manuscript. 

6) Results section

Well written however, a few comments

a. Table 2, last row, second column: what is the difference between the 2 considerations “mother worked for a living” and “mother’s nature of work and where they work”

Response: Thank you for pointing out this unclarity

The two points have been merged to clarify the point of interest and now reads as follows:

Mother working for a living and her nature of work

Table 2, Last column

b. Sentence 163, is that word health or healthy

Response: Thank you for citing this correction.

Word is ``health’’ and not ``healthy’’ 

This has been corrected in the manuscript.

Pg 7, Line 152

7) Discussion

The discussion is mainly full of the findings re-stated with little or no interpretation. 

a. I think it would be good to state the finding briefly, give its explanation or interpretation or meaning and the implication on the baby or mother or family etc and relate to a previous study.

Response Thank you for citing this gap in interpretation of the findings and the guidance provided to improve the discussion. 

The first two paragraphs of the discussion have been rewritten and now read as follows:

We explored the lived experiences of parents to children born preterm to understand the unpriced costs of care parents incur during the first year of life and this is one of the few studies to address the topic. Literature on care and economic implications following preterm birth in our setting is scarce. Available literature is mainly from the developed countries and focused on the health providers’ perceptions. Three themes emerged from the discussions: complex nature of the infant, time to care for the infant and mother as the primary caregiver. 

The complex nature of the infant could have been viewed by majority of the parents as leverage for their infants’ survival. What this meant was that whatever should be done to save the infant should be done irrespective of the cost. Infants’ need for long healthcare during initial hospitalization and after discharge causes anxiety for fear of poor outcome. Parents’ actions during care of their infant are sometimes taken with caution for fear of making mistakes but some decisions lead to large hidden costs which may not be known to the parents. Unlike their term counterparts, the preterm infants require more time for their day-to-day care both during admission and after discharge especially during the first year of life and the mother plays a central role during this time.

Pg 10, Line 222 - 235

8) Objective

Sentences 233 to 234; This is similar to your objective, but not the same.

"Explore and document the indirect costs to parents towards the care of a preterm infant in the first year of life in a low resource setting" this is what you have in the abstract.

Then in the introduction

“The aim of the study was to explore indirect costs parents incur following birth of a preterm infant and document the parents’ experiences on care in the first year of life”

Then in the method section

“We conducted an exploratory qualitative study to explore parents’ views on their experiences regarding the unpriced cost of care of children born preterm and with low birth weight”

i. Based on these, First, you want to explore the indirect costs associated with preterm births, secondly, you wanted to document the parent’s experiences on care in the first year.

I think it’s better for you to decide on the message you want to communicate in relation to pre-term baby care and clearly state the objective and be consistent across the different sections.

Response: Thank you very much for highlighting this discrepancy and offering the guidance. 

The objective has been rephrased and restated in all the sections to communicate same message. The objective has been restated as follows:

In the abstract

From- Explore and document the indirect costs to parents towards the care of a preterm infant in the first year of life in a low resource setting.

To read- We aimed to explore and document the unpriced costs of care parents incur following birth of a preterm infant in the first year of life in a low resource setting.

Pg 1, Line 4 -5 

Introduction section

From- The aim of the study was to explore indirect costs parents incur following birth of a preterm infant and document the parents’ experiences on care in the first year of life

 To read- The aim of the study was to explore and document the unpriced costs of care parents incur following birth of a preterm infant for the first year of life in Kampala, Uganda. 

Pg 2, Line 50 - 52 

Methods section 

From- We conducted an exploratory qualitative study to explore parents’ views on their experiences regarding the unpriced cost of care of children born preterm and with low birth weight

To read- We conducted an exploratory qualitative study to document parents’ views on their experiences regarding the unpriced cost of care of children born preterm in the first year of life. 

Pg 3, Line 57 - 58 

ii. Sentences 237 to 238; I found it a good statement that could add up to the argument or gap in the introduction section.

Response: Thank you very much for this feedback and guidance.

Statement has been used in the introduction section to add up to the gap and reads as follows: 

These estimated costs are majorly healthcare related costs and literature is mostly from high income countries (HIC). There is scanty literature on cost of care for preterm babies and no literature on the indirect cost of care of preterm infants in LRCs.

Pg 2, Line 37 - 39

9) References

You need to use the writing style recommended by PLOS ONE for your references and in-text citation. Kindly refer to Author’s instructions or guide.

Response: Thank you for the recommendation.

This section has been rewritten using the writing style recommended PLOS One and in-text citation. 

Reviewer# 2

1) Abstract:

a. Line 3: what other costs are you referring to, if they are not direct

Response: Thank you for pointing out this unclarity 

The statement has been rephrased 

From The indirect costs related to lost productivity and other costs during the course of care of preterm infants are difficult to estimate

To The costs related to lost productivity to contributions to work during the course of care of preterm infants are difficult to measure.

Pg 1, Line 3 - 4

b. Line 12 -13: i) complex nature of the infant, ii) time to care for the infant, iii) mother as the primary caregiver. The phrasing of these findings, does not make them sound like indirect costs that you set out to explore.

Response: Thank you for the comment about the phrasing used in our study.

This is true they do not sound like the indirect costs we set out to explore. After following your guidance of renaming the indirect costs as unpriced costs, the phrases explain the parents’ experiences in context. And we have thus maintained them in the manuscript and made effort to explain their meaning in the current study.

2) Introduction:

a. This section is not focused on the challenge at hand – indirect costs associated with care for newborns after discharge from the hospital.

i. The problem/gap in knowledge is not clearly articulated in this section.

Response: Thank you for pointing out this concern.

The section has been rewritten to articulate the knowledge gap on the subject of unpriced costs of care parents incur following preterm birth in a low resource setting. 

ii. What have other researchers reported on this subject of indirect costs in low resource settings, beyond mentioning that it is estimated to be 10 times higher than in hospital care.

Response: Thank you for asking this important question.

Available literature in the low resource setting has been on the direct costs which mainly cover initial hospital admission to the time of discharge. We have not found literature on the indirect or unpriced cost of care we are discussing in the current manuscript.

In the current study we mention care of preterm infant as an estimate of 10 times higher than that of an infant born at term. This is a direct cost and not an indirect or unpriced cost. 

3) Please revise this to focus on; what do we know about this issue, what don’t we know about it (gap)?, and what do you want to do about it? (aim of your study)

Response: Thank you for providing this guidance to help us focus.

What do we know about this issue?

The cost of care of a preterm infant in the first year of life is higher than that of an infant born at term. The highest cost of care is spent during initial hospital admission.

What do not we know about this issue?

What we do not know about this issue the unpriced cost of care of a preterm infant in the first year of life. 

What do we want to do about this gap.

The aim of the current this is to explore and document the unpriced cost of care parents incur in the first year of life. The findings from the current study will add to the knowledge gap on the topic of unpriced costs incurred by parents during care of a preterm infant. 

The above revisions have been included in the manuscript to clearly state the gap and aim of the study.

Pg 3, Line 50 - 52

3. Methods

One of the key things missing in this section is the lack of a conceptual/theoretical frame work that has informed this analysis. Relatedly, the Equator guidelines for reporting qualitative research were not followed, this makes this section very uninformative and missing a lot of relevant information.

Response: Thank you for pointing out this omission and offering the guidance.

The section has been rewritten following the COREQ guidelines for reporting qualitative research.

We did not focus on utilizing a particular theoretical or conceptual framework focusing primarily on previously established theory/theories as the basis for our data analysis and interpretation. We based on the notion that while theoretical or conceptual frameworks are helpful in their own context and can bolster the qualitative approach reliance on them in an explorative study like ours begins to limit the ability to see emergent findings in the data and narrow the lens through which the literature and data in the study is viewed. We stand on the premise that the epistemology and methodological rigor used in the conduct of the study provided us with in-depth information and knowledge beyond the theoretical and conceptual framework. 

4. Results

In table 1- please provide an informative and descriptive title that can make it stand alone.

– complete the table margins as well.

Response: Thank you for stating this unclarity and offering guidance to improve the table.

The title for table 1 now reads as follows: 

Socio-demographic characteristics for the study participants. 

The table margins too have been completed. 

Discussion

1.The key message in this section is difficult to discern in the current state, it is probably related to the misalignment between the title, aim and results reported. What should be the take home message for the readers?

Response: Thank you for stating this important observation of the unclarity of the key message to the reader.

 The discussion has been rewritten and aligned with the title, aim and results.

The key message is: 

The results highlight the unpriced cost incurred by the parents through disruption of the work pattern due to the actual and perceived needs of a preterm infant and time to care in a low resource setting. 

This sentence has been added to the manuscript.

Pg 1 & 13, Line 18 – 21 & 292 - 295

2. What theoretical framework underpins the discussion of your results?

Response: Thank you for asking this important question.

We did not consider a particular theoretical framework as highlighted in question 3 above.

i. Line 280 – it should be cost and not course. Relatedly, there are many typos and grammatical errors all over the document that you need to revise.

Response: Thank you for the comment. 

The course here was used to indicate the period of care. 

We have reviewed the manuscript for grammatic errors and typos in the final manuscript. 

6. Conclusion:

a. Line 295- “provides understanding of the unpriced “hidden” costs caregivers experience in context”. I believe that this wording might be a better way to state your aim and title of the study given the results being reported, as opposed to “indirect costs”.

Response: Thank you for providing this guidance.

The term “indirect costs” has been replaced with unpriced cost in the aim, title and entire manuscript to improve and clarify the message being shared in the study.

7. Reference:

1. Number 6 – 11 are incomplete

Response: Thank you for citing this incompleteness.

References have been revised and rewritten to conform to the journal guidelines.

2. Please check all the other references to ensure that they conform to the journal guidelines.

Response: Thank you for taking note of this incompleteness.

All references have been revised to conform to the journal guidelines.

Pg 13 & 14

---

## [Decision Letter · Decision Letter 1]

30 May 2023

PONE-D-22-33576R1“I abandoned my job to look after my baby.” Understanding the unpriced cost of care of a preterm infant: caregivers’ lived experiences.PLOS ONE

Dear Dr. Namiiro,

Thank you for submitting your manuscript to PLOS ONE. After careful consideration, we feel that it has merit but does not fully meet PLOS ONE’s publication criteria as it currently stands. Therefore, we invite you to submit a revised version of the manuscript that addresses the points raised during the review process.

We look forward to receiving your revised manuscript.

Kind regards,

Milton W. Musaba, M.D

Academic Editor

PLOS ONE

Journal Requirements:

Additional Editor Comments:

Dear Dr. Namiiro,

Thank you for revising this manuscript.

Here are some more comments from the reviewers for your consideration.

Please take time to review the grammar in the whole document.

Reviewers' comments:

Reviewer's Responses to Questions

**Comments to the Author**

1. If the authors have adequately addressed your comments raised in a previous round of review and you feel that this manuscript is now acceptable for publication, you may indicate that here to bypass the “Comments to the Author” section, enter your conflict of interest statement in the “Confidential to Editor” section, and submit your "Accept" recommendation.

Reviewer #1: (No Response)

Reviewer #2: All comments have been addressed

2. Is the manuscript technically sound, and do the data support the conclusions?

Reviewer #1: Yes

Reviewer #2: Yes

3. Has the statistical analysis been performed appropriately and rigorously? 

Reviewer #1: N/A

Reviewer #2: Yes

4. Have the authors made all data underlying the findings in their manuscript fully available?

Reviewer #1: Yes

Reviewer #2: Yes

5. Is the manuscript presented in an intelligible fashion and written in standard English?

Reviewer #1: Yes

Reviewer #2: Yes

6. Review Comments to the Author

Reviewer #1: Dear authors,

Thank you for addressing the comments and improving the manuscript.

There are some few areas for improvement observed:

Introduction

“The findings from the current study will add to the knowledge gap on the topic of unpriced costs incurred by parents during care of a preterm infant”.

Having got the knowledge related to unpriced costs incurred by parents during the care of pre-term infants. What the study will inform, is the statement required.

For instance, the findings from this study will enable interventions for the social and financial support of families or parents with preterm infants. You can state it based on the reason you went out to conduct this study. It needs to be an improvement statement.

Methods

Rapport is created, built, or established. You may improve the sentence.

You can remove the note on COREQ from this section. The COREQ is for the consumption of the editorial team and reviewers.

Discussion

“and this is one of the few studies to address the topic”. Please move this to the strength and limitations since it is a strength.

“Literature on care and economic implications following preterm birth in our setting is scarce. The available literature is mainly from the developed countries and focused on the health providers’ perceptions (14-17)”

Let this remain in the introduction if already present.

“Three themes emerged from the discussions: complex nature of the infant, time to care for the infant and mother as the predominant caregiver”. This is a good statement of the findings.

Based on the copy with track changes, I recommend that the whole work in its current state needs editing for grammar and tidying of the sentences. For instance, some phrases could be replaced by words. This will shorten paragraphs and improve flow. I tried in the paragraphs below, you may do it differently.

“The complex nature of the infant could have been viewed by the majority of the parents as leverage for their infants’ survival. This implied whatever should be done to save the infant should be done irrespective of the cost. Infants’ need for long healthcare during initial hospitalization and after discharge causes anxiety for fear of poor outcomes. Parents’ actions during the care of their infant are sometimes taken with caution to avoid making mistakes, however, some decisions lead to large hidden costs that are unknown to the parents. Unlike their term counterparts, preterm infants require more time for their day-to-day care both during admission and after discharge, especially during the first year of life where mothers play a central role.

The small, repetitive, lifesaving roles caregivers perform for their infants, such as the two hourly feeds and kangaroo care, require time and commitment. These may not be easily delegated to a person who is not a nuclear family member or has no prior experience caring for a preterm infant. Also, the bond parents developed during the critical period explains the reason for not trusting another person with the comprehensive care a baby requires. Hence prioritizing care for their infant, although this may disrupt their life.

The time from the birth of a preterm infant to the time they are declared out of danger varies from infant to infant. Similarly, the level of stress, amount of time, resources and effort put into caring for an infant varies between families. Therefore, attaching a cost to these individual aspects of care is difficult because estimating them is hard. Henceforth, it is not surprising the parents viewed the cost of care of preterm infants as expensive since their input is often different. In a previous study, health providers found that the cost of care for infants with lower gestation age and birth weight was higher (10, 18). Physical and emotional drains which may affect performance that were observed in this study have also been reported elsewhere (19, 20). Furthermore, when the mother has to delegate her duties to another person for instance, when resuming work, there was no guarantee that the delegated caregiver would perform exactly as the mother and vice versa. Similar findings are lacking from the literature, hence, the need for more studies in this area. In a nutshell, mothers remain the best caregivers for their preterm babies. Also, flexibility at the workplace or absenteeism to the extremes of having to stop work, all disrupt the life of an individual, their family and community at large”.

Reviewer #2: All the comments were addressed. However, i have 2 minor comments and suggestions that you could include in the manuscript.

1. Can you also include UNCST number in the section of ethical approval?

2. In the recruitment section, at what point did you recruit the mothers and the fathers? Was it after 6 months or a year please clarify?

3. Why were the fathers not included in the FGDs please provide the rationale why you chose IDI's for them.

7. PLOS authors have the option to publish the peer review history of their article (what does this mean?). If published, this will include your full peer review and any attached files.

Reviewer #1: No

Reviewer #2: No

---

## [Author Response · Author response to Decision Letter 1]

30 Jun 2023

30th June 2023

Dear Editor,

RE: Response to Reviewers’ Comments

Thank you to the Reviewers for the comments to improve our manuscript. Below are the point-by-point responses from us. 

Reviewer #1

Dear authors,

Thank you for addressing the comments and improving the manuscript.

There are some few areas for improvement observed:

Introduction

“The findings from the current study will add to the knowledge gap on the topic of unpriced costs incurred by parents during care of a preterm infant”.

Having got the knowledge related to unpriced costs incurred by parents during the care of pre-term infants. What the study will inform, is the statement required.

For instance, the findings from this study will enable interventions for the social and financial support of families or parents with preterm infants. You can state it based on the reason you went out to conduct this study. It needs to be an improvement statement.

Response: Thank you for making this important observation. An improved statement to inform what the knowledge from the study findings will add has been included.

It reads: The findings from the current study will add to the knowledge gap on the topic of unpriced costs incurred by parents during care of a preterm infant. This will guide policies and interventions intended to socially and financially support parents with preterm families. 

Page - 4, Line - 82-83 

Methods

1. Rapport is created, built, or established. You may improve the sentence.

Response: Thank you for this correction. 

The sentence has been improved to read as follows: Before the discussions and interviews started, rapport with the participants was established.

Pg 6, Line 129

2. You can remove the note on COREQ from this section. The COREQ is for the consumption of the editorial team and reviewers.

Response: Thank you for the guidance. The note has been removed from the manuscript.

Discussion

1. “and this is one of the few studies to address the topic”. Please move this to the strength and limitations since it is a strength.

Response: Thank you for the suggestion. This has been moved to strength and limitation section.

Pg - 14, Line - 312-313 

2. “Literature on care and economic implications following preterm birth in our setting is scarce. The available literature is mainly from the developed countries and focused on the health providers’ perceptions (14-17)”

Let this remain in the introduction if already present.

Response: Thank you for this guidance.

Statements have been moved to the introduction section.

Pg - 3, Line - 66- 68

3. “Three themes emerged from the discussions: complex nature of the infant, time to care for the infant and mother as the predominant caregiver”. This is a good statement of the findings.

Response: Thank you for the commendation.

Based on the copy with track changes, I recommend that the whole work in its current state needs editing for grammar and tidying of the sentences. For instance, some phrases could be replaced by words. This will shorten paragraphs and improve flow. I tried in the paragraphs below, you may do it differently.

Response: Thank you for the recommendation.

The entire manuscript has been revised with emphasis to the grammar and replaced some phrases with words as suggested. 

“The complex nature of the infant could have been viewed by the majority of the parents as leverage for their infants’ survival. This implied whatever should be done to save the infant should be done irrespective of the cost. Infants’ need for long healthcare during initial hospitalization and after discharge causes anxiety for fear of poor outcomes. Parents’ actions during the care of their infant are sometimes taken with caution to avoid making mistakes, however, some decisions lead to large hidden costs that are unknown to the parents. Unlike their term counterparts, preterm infants require more time for their day-to-day care both during admission and after discharge, especially during the first year of life where mothers play a central role.

The small, repetitive, lifesaving roles caregivers perform for their infants, such as the two hourly feeds and kangaroo care, require time and commitment. These may not be easily delegated to a person who is not a nuclear family member or has no prior experience caring for a preterm infant. Also, the bond parents developed during the critical period explains the reason for not trusting another person with the comprehensive care a baby requires. Hence prioritizing care for their infant, although this may disrupt their life.

The time from the birth of a preterm infant to the time they are declared out of danger varies from infant to infant. Similarly, the level of stress, amount of time, resources and effort put into caring for an infant varies between families. Therefore, attaching a cost to these individual aspects of care is difficult because estimating them is hard. Henceforth, it is not surprising the parents viewed the cost of care of preterm infants as expensive since their input is often different. In a previous study, health providers found that the cost of care for infants with lower gestation age and birth weight was higher (10, 18). Physical and emotional drains which may affect performance that were observed in this study have also been reported elsewhere (19, 20). Furthermore, when the mother has to delegate her duties to another person for instance, when resuming work, there was no guarantee that the delegated caregiver would perform exactly as the mother and vice versa. Similar findings are lacking from the literature, hence, the need for more studies in this area. In a nutshell, mothers remain the best caregivers for their preterm babies. Also, flexibility at the workplace or absenteeism to the extremes of having to stop work, all disrupt the life of an individual, their family and community at large”.

Pg 10-13, Line: 248 – 314

Reviewer #2 

All the comments were addressed. However, i have 2 minor comments and suggestions that you could include in the manuscript.

1. Can you also include UNCST number in the section of ethical approval?

Response: Thank you for identifying this omission.

The UNCST number was (REF: HS 2690) and has been added to the manuscript.

Pg -7, Line - 161 

2. In the recruitment section, at what point did you recruit the mothers and the fathers? Was it after 6 months or a year please clarify?

Response: Thank you for the comment.

The mothers and fathers were parents of children who were born preterm and the children were between 22 to 38 months at the time of the study.

This was described in the manuscript, methods section under participants and recruitment.

Pg - 7, Line: 112- 113 

3. Why were the fathers not included in the FGDs please provide the rationale why you chose IDI's for them.

Response: Thank you for the query.

Majority of the caregivers registered in our follow-up clinic were mothers. The few fathers we identified could participate at varied times. We were not in position to convene a group of fathers at the same time so we opted to in-depth interviews with them to addressed any unresolved issues. 

We have added this statement to the manuscript to clarify this situation.

Mothers are the primary care givers and for uniformity were used for FGDs, and for any unresolved issues we used IDIs with the fathers who are not readily available in the follow-up clinic. 

Pg – 5, Line: 116- 118

---

## [Editor Report · Decision Letter 2]

24 Jul 2023

PONE-D-22-33576R2“I abandoned my job to look after my baby.” Understanding the unpriced cost of care of a preterm infant: caregivers’ lived experiences.PLOS ONE

Dear Dr. Namiro,

Thank you for submitting your manuscript to PLOS ONE. After careful consideration, we feel that it has merit but does not fully meet PLOS ONE’s publication criteria as it currently stands. Therefore, we invite you to submit a revised version of the manuscript that addresses the points raised during the review process.

ACADEMIC EDITOR: 

See at the end of this email for additional comments that need your attention. 

We look forward to receiving your revised manuscript.

Kind regards,

Milton W. Musaba, M.D

Academic Editor

PLOS ONE

Journal Requirements:

Additional Editor Comments (if provided):

Abstract should be structured

Line 32 – replace “to contributions to” with “at”

Line 40 – is it “thematic approach” or “thematic content approach”? Please cross check

Lines 49 – 50 – this is very far fetched and is not supported by your findings. What can you as the attending clinician start to do now to ameliorate this challenge, even before the policy makers come on board.

Line 89 – should be FGD not just group discussions.

Line 107- participants and recruitment. What was the justification for including newborns below 28 WOA, yet our guideline has just lowered (2021) the age of viability to 26 weeks from 28 weeks?

Line 109 – “children who weighed ≤ 2000 grams at birth and were aged 22-38 months at the time of the study were purposively selected from the clinic registry and invited to participate”. If my interpretation of this statement is correct, it implies that you did not exclusively study the preterm babies, please clarify?

Line 128 – would you consider using “Rapport” and not just relationship?

Line 130 – what was the use of the note taker when you were recording? At what point did you inform your participants about the recording if it done as you mentioned in the abstract.

Line 274 – improve the punctuation and grammar, I wound suggest you add “, which” after baby.
---

## [Author Response · Author response to Decision Letter 2]

31 Jul 2023

31st July 2023

Dear Editor,

RE: Response to Reviewers’ Comments

Thank you to the Reviewers for the comments to improve our manuscript. Below are the point-by-point responses from us. 

Additional Editor Comments 

1. Abstract should be structured

We appreciate this comment from the reviewer. The abstract has been structured as follows: Background, Methods, Findings and Conclusion

Page 2, Lines 31, 35, 41 & 47

2. Line 32 – replace “to contributions to” with “at”

Thank you for correction. This has been done.

Page 2, Line 32

3. Line 40 – is it “thematic approach” or “thematic content approach”? Please cross check

We used the thematic approach.

4. Lines 49 – 50 – this is very far-fetched and is not supported by your findings. What can you as the attending clinician start to do now to ameliorate this challenge, even before the policy makers come on board.

We are grateful to the reviewer for this comment. It actually sounds far-fetched but not impossible when there is good will and planning. At clinician’s level to ameliorate the challenge, initiating the conversation early on financial planning is one approach we are undertaking with the parents. This had been included as a recommendation to the abstract and conclusion and now reads: 

We recommend guidance on financial planning, development of policies and programs on social and financial support for parents and future studies on indirect costs of preterm care.

Page 2, line 49-50 & Pg 13, Line 307-309

5. Line 89 – should be FGD not just group discussions.

We appreciate this correction and it has been revised to read focused group discission.

Page 4, Line 89 

6. Line 107- participants and recruitment. What was the justification for including newborns below 28 WOA, yet our guideline has just lowered (2021) the age of viability to 26 weeks from 28 weeks?

We appreciate the reviewer for pointing out this unclarity. We meant chronological age and not WOA of the participants. To clarify this, chronological age has been added, and the sentence has been revised to read: 

Parents of children who weighed ≤ 2000 grams at birth and were aged 22-38 months, chronological age at the time of the study were purposively selected from the clinic registry and invited to participate.

Page 5, Line 109-110

7. Line 109 – “children who weighed ≤ 2000 grams at birth and were aged 22-38 months at the time of the study were purposively selected from the clinic registry and invited to participate”. If my interpretation of this statement is correct, it implies that you did not exclusively study the preterm babies, please clarify?

We appreciate this comment from the reviewer. Identification of the participants was from the registry and this was followed by a phone call to the caregiver. If their child was admitted or ill at the time of the call, then that child would be excluded. 

To clarify exclusion of infants, this statement has been added: 

Parents whose children were sick or admitted at the time of the study, were excluded. 

Pg 5, Line 111

8. Line 128 – would you consider using “Rapport” and not just relationship?

This has been revised, where “relationship” has been replaced with “rapport.” 

Pg 6, Line 128

9. Line 130 – what was the use of the note taker when you were recording? 

We appreciate this comment from the reviewer. Although recording was being done, the notetaker oversaw this activity and was in charge of keeping the meeting in order.

10. At what point did you inform your participants about the recording if it done as you mentioned in the abstract.

We are grateful for pointing out this unclarity. At the time self-introductions of the participants and research assistants, and study activities were made, mention of the recording was made. To clarify, we have revised the statement to read: 

Introduction of the study objectives and activities were explained to the participants. 

Page 6, Line 128-129

11. Line 274 – improve the punctuation and grammar, I would suggest you add “, which” after baby

We appreciate this suggestion. Punctuation and grammar have been improved. Statement now reads: 

Our study findings indicate care of a preterm infant is generally more costly than care of a term baby which is consistent with other studies.

Page 12, Line 128-129

---

## [Editor Report · Decision Letter 3]

2 Aug 2023

“I abandoned my job to look after my baby.” Understanding the unpriced cost of care of a preterm infant: caregivers’ lived experiences.

PONE-D-22-33576R3

Dear Dr.Namiro,

We’re pleased to inform you that your manuscript has been judged scientifically suitable for publication and will be formally accepted for publication once it meets all outstanding technical requirements.

Kind regards,

Milton W. Musaba, M.D

Academic Editor

PLOS ONE

Additional Editor Comments (optional):

Thank you for responding to all the comments from the reviewers.
---

## [Editor Report · Acceptance letter]

8 Aug 2023

PONE-D-22-33576R3 

“I abandoned my job to look after my baby.” Understanding the unpriced cost of care of a preterm infant: caregivers’ lived experiences.  

Dear Dr. Namiiro:

I'm pleased to inform you that your manuscript has been deemed suitable for publication in PLOS ONE. Congratulations! Your manuscript is now with our production department. 

Kind regards, 

on behalf of

Dr. Milton W. Musaba 

Academic Editor

PLOS ONE